**Subject Category:**
Biology (whole organism)

behaviour/ecology/civil engineering

sand, suction, crab, habitat selection, burrowing

**Author for correspondence:**
Shinji Sassa
e-mail: sassa@p.mpat.go.jp

# Suction-induced habitat selection in sand bubbler crabs

## Shinji Sassa and Soonbo Yang

Port and Airport Research Institute, National Institute of Maritime, Port and Aviation Technology, 3-1-1 Nagase, Yokosuka, Japan

 SS, 0000-0002-0269-200X

We show that a decapod crustacean, the sand bubbler crab (SBC) *Scopimera globosa*, uses suction, which is the tension of moisture in the sediment, to select habitats at normal times and at the time of disaster events, through a range of controlled laboratory experiments and field observations at various sandflats in Japan. When SBCs are released on fields with no spatial suction gradient, their direction of movement is random. However, the situation clearly changes with increasing suction gradients, in which case the SBCs move to suitable zones for burrowing. Predictions based on suction–burrowing relationships coupled with the knowledge of geophysical state changes induced by suction dynamics are consistent with the observed formation of habitats throughout the seasons. Such suction-induced habitat selection in SBCs manifests itself in a robust way even following sudden events such as typhoons, where erosion and deposition processes distinctly alter the geomorphological profiles, as well as the states of suction, yet consistently yielding habitats at the newly formed, suitable suction environments. Repeated battles were observed in a suitable suction environment over burrows, with the competition rate more than seven times as high as that in a critical suction environment for burrowing.

## 1. Introduction

Capillarity, or surface tension of water, as noted very early by Newton, has been one theme in physics that has demanded quantitative understanding for the last 200 years [1] and is relevant to diverse biological phenomena in a wide range of species [2].

Our experience with building sandcastles or digging tunnels on a sandy beach readily tells us the power of suction [3,4] that represents the tension of pore water between sand grains when they get wet, yet are not submerged. Suction produces effective cohesion in sand that has no inherent cohesion, allowing burrows to stand by themselves [5]. The dynamics of such suction, associated with the tide- and wave swash-induced fluctuations in

groundwater level, plays a substantial role in controlling the geophysical environment of habitats. Namely, the hardness of the surface intertidal sediments varies by a factor of 20–50 due to suction development and suction-induced sediment compaction in the essentially saturated states of intertidal flats and beaches [6,7]. The suction dynamics-induced compaction and associated variations in shear strength of the sediments have also been shown to play a crucial role in intertidal flat sandbar morphodynamics [8] and in forming the intertidal flat stratigraphy of sandy, muddy and sand–mud layered sediments [9]. In addition, it has been found that the suction governs suitable and critical conditions for the burrowing of diverse species inhabiting intertidal flats and sandy beaches [5,7,10].

Notably, for sand bubbler crabs (SBCs) *Scopimera globosa*, there exist optimum and critical suction conditions for burrow development [5]. That is, suction allows burrowing and governs the depth to which burrowing is physically possible; however, increasing suction makes burrowing more difficult due to the increased strength, yielding optimum (OP) and critical (CR) suction environments.

The present study explored the SBCs' potential to select habitats at suitable suction environments at normal times and at the time of disaster events, through a range of controlled laboratory experiments and field observations.

## 2. Material and methods

Suction, $s$, means the tension of moisture in the sediment [3] and is defined by

$$s = u_a - u_w, \tag{2.1}$$

where $u_a$ is the atmospheric air pressure, and $u_w$ is the pore water pressure in the sediment. By definition, suction is equal to zero at the groundwater level.

The void state of the sediment is represented by void ratio $e$, which is related to the sediment porosity $n$,

$$e = \frac{n}{1 - n}. \tag{2.2}$$

The state of sediment packing, such as dense or loose, can be denoted by the sediment relative density $D_r$,

$$D_r = \frac{e_{max} - e}{e_{max} - e_{min}}. \tag{2.3}$$

For a given sediment, the maximum void ratio $e_{max}$ represents the loosest possible packing, and the minimum void ratio $e_{min}$ represents the densest possible packing [11]. Thus, the $D_r$ value is a normalized index by which to assess the packing states of sandy sediments.

Laboratory experiments were conducted to see how SBCs respond to situations where a spatial suction gradient is prescribed and varied, and how burrows of SBCs are formed in a given suction gradient, by using a two-dimensional flume shown in electronic supplementary material, figure S1. Sand deposits with a prescribed relative density $D_r = 60\%$ were repeatedly formed in the flume by using the intertidal sandy sediments taken from the Nojima sandflat ([7], electronic supplementary material, figure S6). Suctions $s$ and suction gradients $ds/dx$ at the level of the sediment surface were prescribed and varied by controlling the water levels in the reservoir on both sides of the flume, which changed the groundwater levels in the sediment. In the essentially saturated states where the developed suction is below the air-entry suction [4], a linear relationship holds true between suction $s$ and groundwater level GWL such that $s = -\gamma_w \cdot$ GWL, where $\gamma_w$ is the submerged unit weight of water [6]. In all experiments, the air temperature, the water temperature and the salinity of the water and pore water were kept essentially constant at 20–21°C, 19–20°C and 27 psu, respectively. Prior to the experiments, SBCs were maintained in the laboratory under aerated fresh seawater in the intertidal sediments for over one month to ensure that any endogenous physiological rhythms were abolished [12]. SBCs were placed one by one on the sand surface under prescribed suction conditions to observe the direction of movement (figure 1). Experiments were also carried out to see how burrows of SBCs are formed in a given suction gradient. The spatial suction gradient $ds/dx$ was set equal to 2.8 kPa m$^{-1}$. SBCs were placed one by one on the sand surface under critical (CR) and optimum (OP$^A$ and OP$^B$) suction conditions, respectively (figure 2 and electronic supplementary material, figures S2–S5). The photomicrograph of the distal zone of the crab's leg (figure 1*a*), being inserted in the meniscus of the capillary water between sand grains on the surface was taken using a digital microscope VHX-1000 (Keyence, Inc.). A total of 623 individuals were used in the laboratory experiments. The carapace widths and wet weights of the crabs were 8 ± 1 mm and 0.37 ± 0.1 g, respectively.

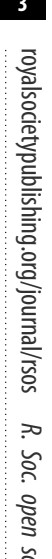

**Figure 1.** Suction-induced displacement in sand bubbler crabs. (*a*) Photomicrograph of the distal zone of the leg, being inserted in the meniscus of the capillary water between sand grains on the surface. The exposed sands are in an essentially saturated state. The scale is shown. (*b*) Direction of movement on given spatial gradients of capillary suction [3] at the same critical suction (CR). The critical suction was equal to 2.1 kPa, which made burrowing difficult, with limited burrow development [5], due to increased hardness of the sand. The optimum suction (OP) was equal to 1 kPa, which realized optimum conditions for burrow development [5]. The suction *s* denotes that at the level of the sand surface and was prescribed by controlling the groundwater levels (GWLs) in a two-dimensional flume (electronic supplementary material, figure S1). In essentially saturated states below air-entry suction [6], a linear relationship between *s* and GWL holds such that $s = -\gamma_w \cdot \text{GWL}$, where $\gamma_w$ is the submerged unit weight of water. The carapace widths of the crabs observed were $8 \pm 1$ mm. (*c*) Proportion of individuals moving to the optimum suction environment versus gradient of suction, after being placed on the same critical suction region. The detection capabilities steadily increased with increasing spatial suction gradient. All experiments were conducted by placing individual crabs one by one on the sand surface under prescribed suction conditions. Error bars indicate s.e.

Field observations were performed at four intertidal sandflats: Naha, Isumi and Nagura Amparu sandflats and Tokuyama artificial sandflat, as shown in electronic supplementary material, figure S6. The density surveys of SBCs, instantaneous measurements of suction and groundwater level, and *in situ* undisturbed sediment samplings were conducted at Naha, Isumi and Nagura Amparu sandflats during spring low tides in the period from 2009 to 2015 with different seasons (figure 3). Suctions were measured using tensiometers [5–7]. The device consisted of a plastic tube with a porous ceramic cup and was filled with de-aerated water. The crab densities were measured by setting four square units of 200–500 mm at each station of the geophysical measurements stated above to count the number of burrows excavated by SBCs. Geomorphological profiles together with the groundwater levels during spring low tides were measured 7 days following the passage of typhoon no. 6 (May 2015) and 5 days following the passage of typhoon no. 15 (August 2015), at the Naha sandflat (electronic supplementary material, figure S6), using a Global Navigation Satellite System (Trimble R4 GNSS, Trimble, Inc.).

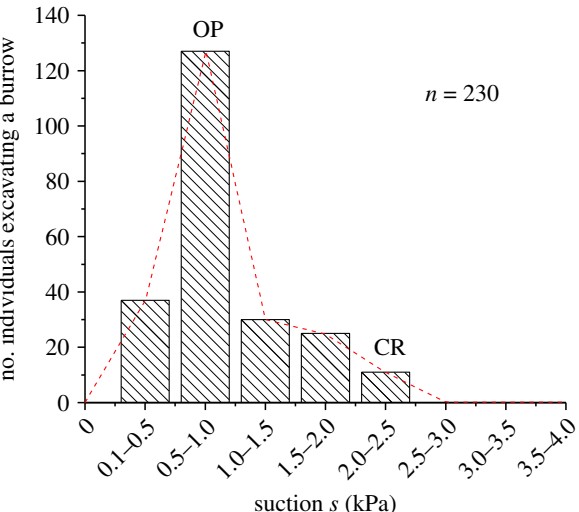

**Figure 2.** Summary of experiment results for suction-induced habitat selection in sand bubbler crabs. Number of individuals excavating a burrow versus suction $s$. This figure shows the sum of the number of sand bubbler crabs excavating a burrow versus suction presented in figures S3–S5, respectively. The majority of individuals performed burrowing activities at the optimum (OP) suction environment and in the vicinity.

Continuous integrated observations of suctions and the densities of the individual crabs were performed at six stations in the along-shore and cross-shore directions of this flat (figure 4). At the Tokuyama artificial sandflat (electronic supplementary material, figure S6), continuous measurements of suction and video (GZ-RX500-B: JVC, Inc.) recordings of the SBCs' behaviours were performed during periods of exposure on 28 September 2015 and 21 July 2016, respectively, at five cross-shore stations, where each monitoring area was $0.84 \times 0.47$ m, to observe the process of displacement, new burrow excavation and the occurrence of competition and battles between crabs over the burrows (figure 5).

# 3. Results and discussion

## 3.1. Laboratory experiments using the two-dimensional flume

A series of experiments were performed to see how SBCs respond to situations where a spatial suction gradient was prescribed and varied in a field, by using the two-dimensional flume in which groundwater level variations could be controlled (electronic supplementary material, figure S1). When SBCs were placed on sands in exposed yet essentially saturated states, they placed the tip of each leg in the meniscus of the capillary waters between sand grains (figure 1a). Since force-sensitive mechanoreceptors are concentrated in the distal zones of the legs [13], this suggests that SBCs sensed capillary suction. In an OP environment, SBCs immediately started digging (burrowing). However, in a CR environment with no spatial suction gradient, namely, where the same critical suctions were prescribed at and around the SBCs, they did not start burrowing but instead exhibited random movements, with essentially identical numbers of individuals moving leftward and rightward (figure 1b). By contrast, in a CR environment where the suction gradient was not zero but spatially varied, their responses changed distinctly, with the majority of individuals moving toward the OP environment (figure 1b). The proportion of individuals moving to the OP side increased steadily with increasing suction gradient, $ds/dx$, and the proportion exceeded 90% when $ds/dx$ was equal to $2.8$ kPa m$^{-1}$ (figure 1c). Indeed, the experimental results shown in figure 2 demonstrate that the majority of the individuals performed burrowing activities in the OP environment and in the vicinity.

## 3.2. Suction-induced habitat selection in the field

A question now arises as to how the habitat is formed in the field, particularly regarding suction. The density of individuals versus sediment suction was recorded in the intertidal sandflat Isumi (figure 3a). There was an apparent peak in the density of SBCs at a suction of approximately 1 kPa.

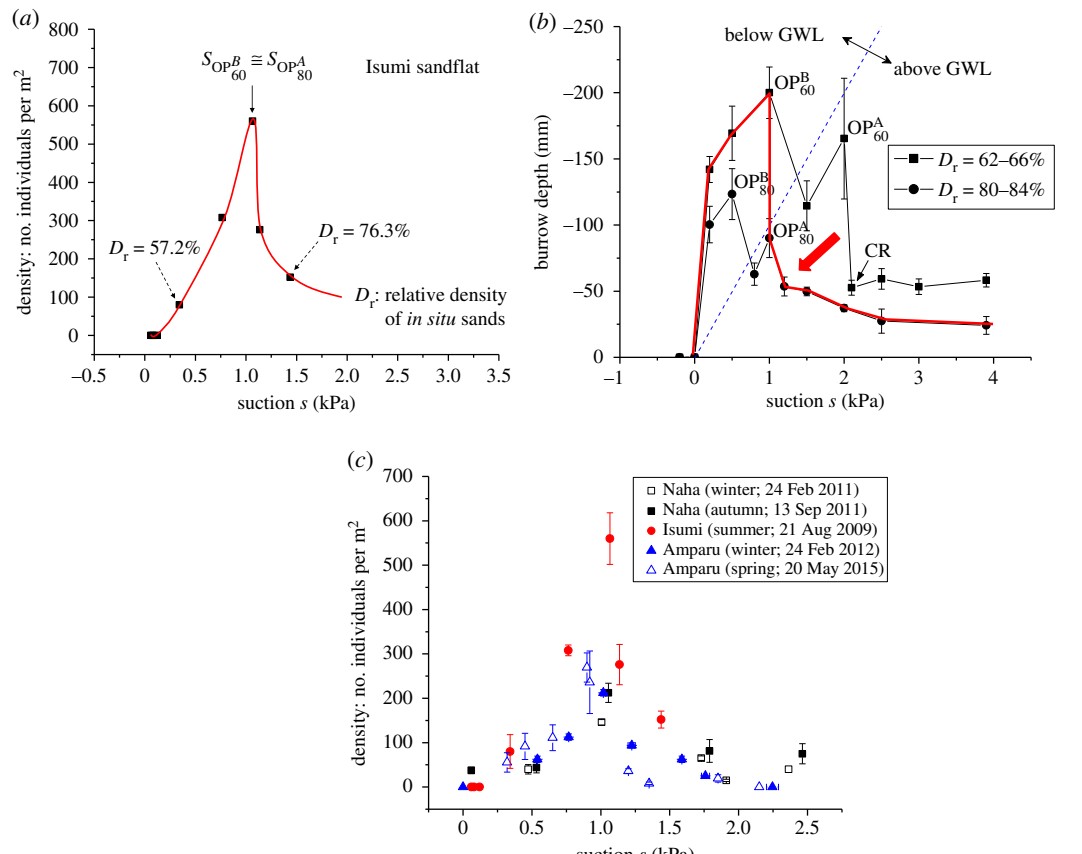

**Figure 3.** Suction-induced habitat selection in the field. (*a*) Density of individuals versus suction *s* in the Isumi sandflat (electronic supplementary material, figure S6). Observations were performed at a spring low tide. $D_r$ denotes relative density, which is a normalized index for assessing loose or dense states of sands, obtained by *in situ* undisturbed sampling and laboratory soil tests. The symbols $s_{OP_{60}^B}$ and $s_{OP_{80}^A}$ denote optimum suction conditions, as described in (*b*). (*b*) Burrow development versus suction *s* as obtained by laboratory experiments [5]. Sand compaction induced by suction dynamics under tide-induced groundwater level fluctuations [6] is superimposed, showing the expected suction–burrowing relationships in the field, denoted in red. For a given relative density $D_r$, there existed two optimum conditions, $OP^A$ and $OP^B$, for burrowing above and below the GWL, respectively. The suction dynamics-induced sand compaction above the GWL gave rise to the condition $s_{OP_{60}^B} \cong s_{OP_{80}^A}$, which appears in (*a*). (*c*) Density of individuals versus suction *s* at the Naha, Isumi and Nagura Amparu sandflats (electronic supplementary material, figure S6) in different seasons. Distinct habitats were formed at suctions around 1 kPa throughout different flats and seasons. Error bars s.e.

Below and above this suction, the relative density, $D_r$, which is a normalized index representing loose or dense states of sands, varied markedly, with denser sands at higher suction. This habitat selection is consistent with the suction–burrowing relationships combined with the knowledge of geophysical state changes induced by suction dynamics [6], as shown in figure 3*b*. Namely, with reference to a conceptual model (electronic supplementary material, figure S2), the presence of suction, $s > 0$, at the level of the sand surface allowed burrow development, which corresponded to two distinctive regions below and above the groundwater level (GWL). After reaching the GWL, the SBCs descended further by using a closed cavity containing entrapped air and the crab itself, since there was no suction present below the GWL. This yielded two optimum conditions, $OP^B$ below the GWL and $OP^A$ above the GWL for a given relative sand density $D_r$ (electronic supplementary material, figure S2). It is important to remark that the dynamics of suction with a magnitude over 1 kPa, associated with tide-induced GWL fluctuations, gave rise to marked sand compaction [6], giving a 20% or higher increase in $D_r$. Such suction dynamics-induced cyclic sediment compaction is superimposed on the suction–burrowing relationships, leading to the envelope in red (figure 3*b*), indicating a unique optimum suction environment for burrowing. This suggests that a distinct habitat may be formed around a suction equal to 1 kPa on the basis of the suction-induced displacement described above. Indeed, the prediction accounted for the observed habitat selection in the field (figure 3*a*), which is further reinforced by the results of laboratory observations (figure 2). The habitat selection at suitable suction environments was observed at different sandflats and through different seasons (figure 3*c*).

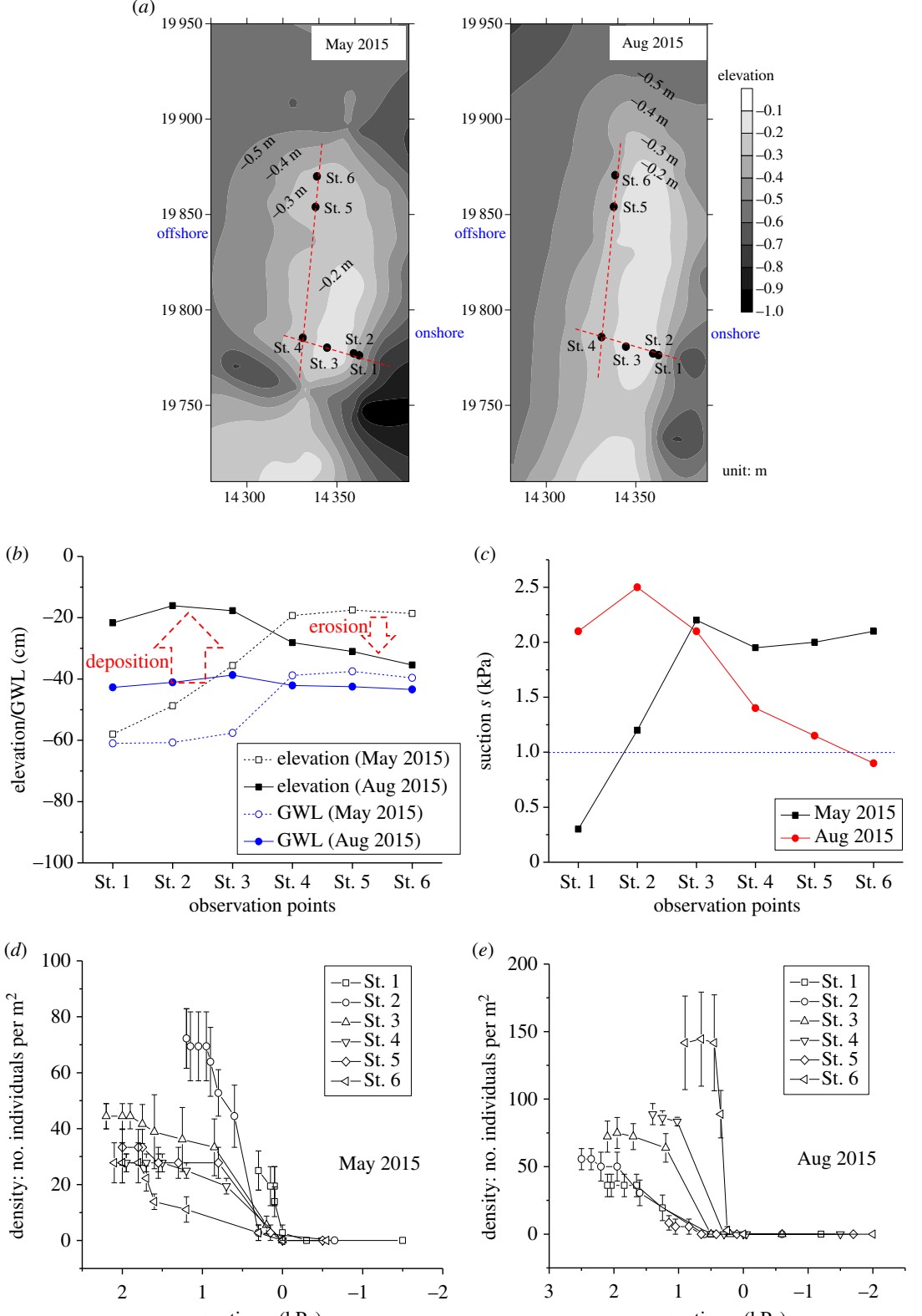

**Figure 4.** Robustness of suction-induced habitat selection following sudden events. (*a*) Geomorphological profiles following the passage of typhoon no. 6 (May 2015) and typhoon no. 15 (August 2015), at the Naha sandflat (electronic supplementary material, figure S6). Locations of the six observational stations in the along-shore and cross-shore directions are shown. (*b*) Changes in the ground elevations and groundwater levels (GWLs) showing erosion and deposition patterns. (*c*) Changes in the spatial suction states. Suction denotes a maximum value developed at each observation point during the spring low tide. The position of the suitable suction environment, $s \cong 1$ kPa, shifted distinctly. (*d,e*) Density of individuals versus suction $s$ relationship following each of the typhoons no. 6 and 15. Temporal changes are shown. Error bars indicate s.e.

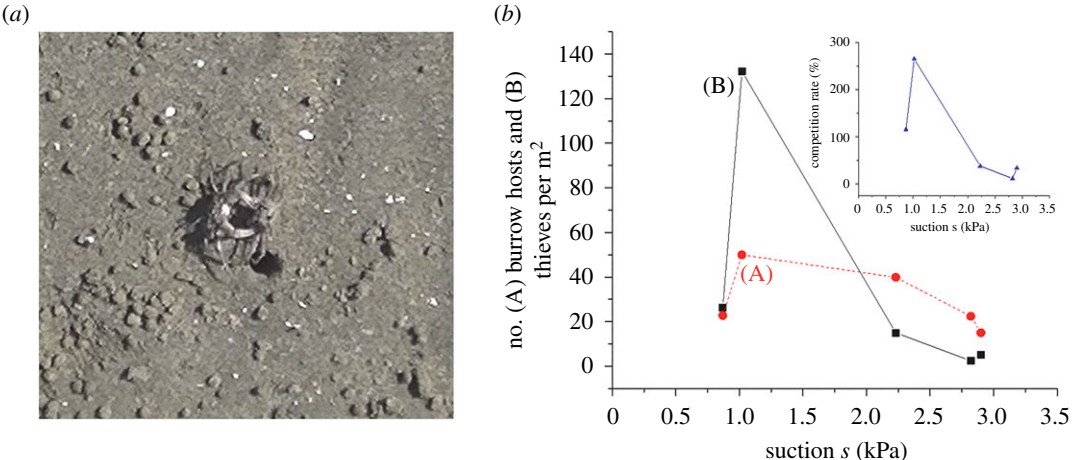

**Figure 5.** Stealing of burrows and competition in a suitable suction environment. (*a*) Photograph showing the occurrence of a battle between two crabs, i.e. a thief and a host, over the burrow occupied by the host. Continuous video recordings/observations were performed at the Tokuyama artificial sandflat (electronic supplementary material, figure S6) during a spring low tide. (*b*) Number of (A) burrow hosts and (B) thieves per m$^2$ versus suction *s*. Suction denotes a maximum value developed during the spring low tide at each observational station. The competition rate represented by (B)/(A) is also shown.

## 3.3. Suction-induced habitat selection following typhoon events

Observations of the benthic- and geo-dynamics were performed several days after each of two typhoons that passed an intertidal sandflat (Naha, electronic supplementary material, figure S6), accompanied by prior observations taken during ordinary periods. Six observational stations were set in the cross-shore and along-shore directions in a $150 \times 50$ m rectangle (figure 4*a*). Following the passage of the two typhoons, typhoons no. 6 and 15 which occurred in 2015, the geomorphological profiles changed markedly, such that the offshore sides of the flat (Sts. 4–6) were eroded, where the coral base was exposed in part (St. 5), leading to deposition at the onshore sides (Sts. 1–3). Accordingly, the GWLs varied, with shallower GWL at the offshore sides and deeper GWL at the onshore sides from the sediment surface (figure 4*b*). As a consequence, the spatial profiles of suction changed distinctly; the suitable suction environment around $s = 1$ kPa shifted from Station 2 to Station 6 (figure 4*c*). Figure 4*d*,*e* reveals that a distinct habitat with a peak density of SBCs was observed at Station 2 7 days following the passage of typhoon no. 6, and at Station 6 5 days following the passage of typhoon no. 15. These findings clearly indicate the SBCs' swift and robust ability to select habitats at a suitable suction environment even following sudden events.

## 3.4. Crab competition for burrows

Continuous video recordings of the displacement at an artificial sandflat (Tokuyama, electronic supplementary material, figure S6) revealed an interesting behaviour of SBCs. As described above, a distinct habitat was formed at a suction around 1 kPa. While incoming crabs excavated new burrows, which constituted about 10% of the total number of burrows, in suitable suction environments, the majority of the crabs stole those burrows by engaging in battles with the burrow hosts, as shown in figure 5*a*. The number of thieves was 132 over 50 burrow hosts per m$^2$, meaning that, on average, about two and a half turnovers of burrow hosts took place in battles for the same single burrow (figure 5*b*). The number of thieves declined rapidly with increasing suction with no new burrows excavated at suctions exceeding 2 kPa. As a consequence, the competition rate in the suitable suction environment reached 265%, which is more than seven times as high as that in the critical suction environment.

## 4. Conclusion

Overall, the above results demonstrate the suction-induced habitat selection in SBCs, which was robust at all times under ordinary and abruptly changing environments. The discovery of this new form of habitat selection by animals will broaden the intersection between ecology and geophysics. As the capillary

suction responds sharply to changes in groundwater levels driven by climate change [14], this may also affect conservation strategies of diverse creatures [15] in changing geoenvironments.

Data accessibility. Data available from the Dryad Digital Repository: https://doi.org/10.5061/dryad.71c14bt [16].
Authors' contributions. S.S. conceived and designed research. S.S. and S.Y. performed the field and laboratory work, analysed the data and wrote the manuscript. Both authors gave final approval for publication.
Competing interests. We declare we have no competing interests.
Funding. This research was supported by the Japan Society for the Promotion of Science Grants-in Aids for Scientific Research (JP15H02265 and JP20360216).
Acknowledgement. We would like to thank Yoichi Watabe for discussion in the early stage of this research. We also thank Keita Watanabe for assistance during fieldwork.

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
