## [Reviewer comments · Royal Society Open Science]

Review History

RSOS-180976.R0 (Original submission)

Review form: Reviewer 1 (Felicita Scapini)

Is the manuscript scientifically sound in its present form?

Yes

Are the interpretations and conclusions justified by the results?

No

Is the language acceptable?

Yes

Is it clear how to access all supporting data?

Yes

Do you have any ethical concerns with this paper?

No

Have you any concerns about statistical analyses in this paper?

No

Recommendation?

Major revision is needed (please make suggestions in comments)

Comments to the Author(s)

The study on the response of sand bubbler crabs to sand-suction dynamics is highly interesting, well conducted and worth of publication. Still, the interpretation of the results and text layout need a thorough revision, as indicated below. The “detection of sediment suction state” was not analysed, but simply inferred from the observation of behaviour and literature. “Migration” was not observed, but the orientation to suction gradient was tested in laboratory experiments, while zonation and zonal recovery after typhoon events were analysed in the field. “Habitat formation” is a general concept: here burrowing/burrow development/competition for burrows were studied. Throughout the text, the terms “environment” and “habitat” are used with different meanings. I suggest to use “beach zone”, defined by specific features, including sediment suction; “habitat”, when referring to the zone where the SBC live; “burrow”, the constructions by the SBCs.

Title – it is misleading, as no detection (of what?) nor migration were observed. I suggest a new title that would better reflect the study: “Suction-induced habitat selection, orientation and burrowing in sand bubbler crabs”.

Abstract – Please, add the scientific name of the studied species, which is a crustacean decapod. In general the term “invertebrates” should be avoided because it refers to very different animal groups. Please, mention the environment under study, i.e. sand flats, and geographically locate the study area; specify that both laboratory experiments and field observation were carried out; lines 7, 11: change “environments” with “beach zones”;

line 8: change “form habitats” with “develop burrows”; explain what “normal times” and “times of events” are;

line 13: it is not clear;

line 14: Please, change “detection, migration and habitat formation” with “Suction-induced habitat selection, orientation and burrowing in sand bubbler crabs”, as suggested above;

line 18: please change the sentence, as an environment can not “give rise”;

line 19: please define “critical suction environment”;

lines 20-21: Delete the sentence, as this is a naïve statement, which would need a forum discussion, as the drivers of human habitat selection behaviour are much more complicated than crabs’ ones. Sediment suction/soil characteristic as drivers for human constructions are beyond the aims of the study!

Keywords: change “migration” with “orientation”

Introduction

Lines 27-28 – This is a naïve statement that ignores the huge literature on orientation to cues of different nature, gradients and directional stimuli, on humans and animals including invertebrates; I suggest to delete the whole sentence and start with the keyword “Capillarity”.

Lines 31-32 – Please delete “from insects to humans” (no direct relationships, ignoring marine and interstitial animals...); quote a review paper on the subject, which may also include plants, which are biological objects highly influenced by capillarity;

Line 34 – “magical power” is not a scientific concept and should be avoided;

Line 39 – change “surficial” with “surface”;

Line 48 – please, add here the scientific name of the species studied; define “critical suction”;

Line 53-54 – see above

Methods

Line 73- change “habitats” with “burrows”;

Lines 77, 92-93 – please geographically locate the study sites;

Line 82 – How many crabs were tested? At what time of day? What was the treatment before and after the experiments (captivity time, feeding, handling, were individuals re-used?);

Line 84 – change with “Experiments were carried out to see how burrows are formed by sand

bubbler crabs”;

Line 91 – “equal to 10mm”: this is not possible! Kindly write the error or write “around 10mm”;

Line 94 – “density surveys”: add here the sampling method and design used, which is specified at lines 166-169;

Line 97 – “in different seasons”: kindly specify the differences between seasons for international readers; “tensiometers”, please specify the characteristics of the instruments used;

Lines 110-111 – change “process of migration” with “displacement”; change “creation” with “excavation”;

Results and discussion – The results, discussion and conclusions should be clearly separated. I suggest to add subtitles instead of repetitions of the methods.

Lines 115-118 – replace with a subtitle, referring to figure S1, e.g., “Laboratory experiments using the two dimensional flume”;

Lines 120-122 – “Since...suction”: please remove to the Discussion, as this was not tested, but was inferred from the literature;

Lines 131-132 – “These results...environments” should be removed to the Conclusions;

Lines 133-135 – replace “A question...suction” with a subtitle like “Suction-induced burrow excavation in the field”; Replace “One such...figure 3a” with “The density of individuals versus sediment suction was recorded in the intertidal sandflat Isumi”.

Lines 138 – Change “habitat formation” as explained above;

Line 142 – Change “corresponds” with “corresponded”;

Lines 145-150 – This long sentence is not clear;

Line 154 – Change “habitat” with “burrow”;

Line 156 – Change “habitat formation” with “burrowing activity”;

line 159 – “more suitable optimum” sounds odd; you may simply write “above/below G.W.L. OP”;

Lines 161-162 – Please change “formation of habitats” with “excavation of burrows”;

“manifested” is a transitive verb and should be “manifested itself”, better “was observed”;

Lines 164-165 – The statement belongs to the Conclusions;

Lines 166-169 – The sentence is a repetition of the Methods and should be replaced by a subtitle, like “Suction-induced burrowing behaviour following typhon events”;

Line 178 – “manifested” is a transitive verb and should be “manifested itself”, better “was observed”;

Lines 179-181 – The statement belongs to the Conclusions;

Line 181 – Add here a subtitle like “Crab competition for burrows”;

Line 187 – change “numbered” with “The number of thieves was 132”;

Line 190 – Change “created” with “excavated”;

Lines 193-194 – “These findings... society”, see comment for lines 20-21; in any case the statement belongs to the Discussion;

Line 195 – Start here the “Conclusions” section, taking into account the above comments. Please, note that the analysis of burrow excavation and defence is not new; what is new is the orientation with respect to a suction gradient, i.e., a geophysical gradient.

The reviewer: Felicita Scapini, Department of Biology, University of Florence, Italy

Review form: Reviewer 2

Is the manuscript scientifically sound in its present form?

Yes

Are the interpretations and conclusions justified by the results?

Yes

Is the language acceptable?

Yes

Is it clear how to access all supporting data?

Yes

Do you have any ethical concerns with this paper?

No

Have you any concerns about statistical analyses in this paper?

I do not feel qualified to assess the statistics

Recommendation?

Major revision is needed (please make suggestions in comments)

Comments to the Author(s)

The findings of this paper show that the sand bubbler crab (SBC) utilizes suction to detect and migrate to suitable environments at different times. These results also show a suction-induced detection, migration and habitat formation by SBC. The authors state that this is a new form of environmental detection and migration by animals. The capillary suction responds to changes in groundwater levels driven by climate change. These results may also affect conservation strategies.

General comments:

1. English needs review. For example, 'A series of experiments was performed...' Line 115. Change "was" by "were". Review the entire paper for additional edits.
2. Use the acronym SBC in the text as appropriate. There are several sections where the words "sand bubbler crab" could be substituted with SBC. Lines 72, 74, 82, 84, etc.
3. Use past tense to describe aims, methods, and results. For example: "The present study aims to explore...". Change it by "The present study explored the potential of the SBC..." Line 53.

Major comments:

Abstract:

1. What do the authors mean by "better living conditions"? Line 20

Results and Discussions:

1. What is a "suitable suction environment"? Line 132
2. Please explain the logic for the following statement: "a distinct habitat may be formed". Line 154
3. Please explain the statement in line 193-194 "these findings indicate..., as in human society". Are the authors comparing the behavior of SBC with human behavior? If so, what variables of human behavior were compared with SBC behavior?
4. How exactly do the authors suggest that their findings will "broaden the intersection between biology, physics and engineering"?

The findings of this paper are very interesting but could be strengthened by avoiding assumptions rather than the results provided for the experiments and field observations.

Decision letter (RSOS-180976.R0)

02-Nov-2018

Dear Dr Sassa:

Manuscript ID RSOS-180976 entitled "Suction-induced detection, migration and habitat formation in sand bubbler crabs" which you submitted to Royal Society Open Science, has been reviewed. The comments from reviewers are included at the bottom of this letter.

In view of the criticisms of the reviewers, the manuscript has been rejected in its current form. However, a new manuscript may be submitted which takes into consideration these comments.

Please note that resubmitting your manuscript does not guarantee eventual acceptance, and that your resubmission will be subject to peer review before a decision is made.

Your resubmitted manuscript should be submitted by 02-May-2019. If you are unable to submit by this date please contact the Editorial Office.

Please note that Royal Society Open Science will introduce article processing charges for all new submissions received from 1 January 2018. Charges will also apply to papers transferred to Royal Society Open Science from other Royal Society Publishing journals, as well as papers submitted as part of our collaboration with the Royal Society of Chemistry (<http://rsos.royalsocietypublishing.org/chemistry>). If your manuscript is submitted and accepted for publication after 1 Jan 2018, you will be asked to pay the article processing charge, unless you request a waiver and this is approved by Royal Society Publishing. You can find out more about the charges at <http://rsos.royalsocietypublishing.org/page/charges>. Should you have any queries, please contact openscience@royalsociety.org.

on behalf of Dr Safi Darden (Associate Editor) and Kevin Padian (Subject Editor)
openscience@royalsociety.org

Editor comments:

This is clearly an interesting and well executed study. However, the reviewers found a great many problems with it, although most were minor and are easily fixed. I am recommending a

"reject/resubmit" decision because it will give you more time to revise than a "major revision" decision. Please make sure to address all comments.

Reviewers' Comments to Author:

Reviewer: 1

Comments to the Author(s)

The study on the response of sand bubbler crabs to sand-suction dynamics is highly interesting, well conducted and worth of publication. Still, the interpretation of the results and text layout need a thorough revision, as indicated below. The "detection of sediment suction state" was not analysed, but simply inferred from the observation of behaviour and literature. "Migration" was not observed, but the orientation to suction gradient was tested in laboratory experiments, while zonation and zonal recovery after typhoon events were analysed in the field. "Habitat formation" is a general concept: here burrowing/burrow development/competition for burrows were studied. Throughout the text, the terms "environment" and "habitat" are used with different meanings. I suggest to use "beach zone", defined by specific features, including sediment suction; "habitat", when referring to the zone where the SBC live; "burrow", the constructions by the SBCs.

Title - it is misleading, as no detection (of what?) nor migration were observed. I suggest a new title that would better reflect the study: "Suction-induced habitat selection, orientation and burrowing in sand bubbler crabs".

Abstract - Please, add the scientific name of the studied species, which is a crustacean decapod. In general the term "invertebrates" should be avoided because it refers to very different animal groups. Please, mention the environment under study, i.e. sand flats, and geographically locate the study area; specify that both laboratory experiments and field observation were carried out;

lines 7, 11: change "environments" with "beach zones";

line 8: change "form habitats" with "develop burrows"; explain what "normal times" and "times of events" are;

line 13: it is not clear;

line 14: Please, change "detection, migration and habitat formation" with "Suction-induced habitat selection, orientation and burrowing in sand bubbler crabs", as suggested above;

line 18: please change the sentence, as an environment can not "give rise";

line 19: please define "critical suction environment";

lines 20-21: Delete the sentence, as this is a naïve statement, which would need a forum discussion, as the drivers of human habitat selection behaviour are much more complicated than crabs' ones. Sediment suction/soil characteristic as drivers for human constructions are beyond the aims of the study!

Keywords: change "migration" with "orientation"

Introduction

Lines 27-28 - This is a naïve statement that ignores the huge literature on orientation to cues of different nature, gradients and directional stimuli, on humans and animals including invertebrates; I suggest to delete the whole sentence and start with the keyword "Capillarity".

Lines 31-32 - Please delete "from insects to humans" (no direct relationships, ignoring marine and interstitial animals...); quote a review paper on the subject, which may also include plants, which are biological objects highly influenced by capillarity;

Line 34 - "magical power" is not a scientific concept and should be avoided;

Line 39 - change "surficial" with "surface";

Line 48 - please, add here the scientific name of the species studied; define "critical suction";

Line 53-54 - see above

Methods

Line 73- change "habitats" with "burrows";

Lines 77, 92-93 - please geographically locate the study sites;

Line 82 – How many crabs were tested? At what time of day? What was the treatment before and after the experiments (captivity time, feeding, handling, were individuals re-used?);

Line 84 – change with “Experiments were carried out to see how burrows are formed by sand bubbler crabs”;

Line 91 – “equal to 10mm”: this is not possible! Kindly write the error or write “around 10mm”;

Line 94 – “density surveys”: add here the sampling method and design used, which is specified at lines 166-169;

Line 97 – “in different seasons”: kindly specify the differences between seasons for international readers; “tensiometers”, please specify the characteristics of the instruments used;

Lines 110-111 – change “process of migration” with “displacement”; change “creation” with “excavation”;

Results and discussion – The results, discussion and conclusions should be clearly separated. I suggest to add subtitles instead of repetitions of the methods.

Lines 115-118 – replace with a subtitle, referring to figure S1, e.g., “Laboratory experiments using the two dimensional flume”;

Lines 120-122 – “Since...suction”: please remove to the Discussion, as this was not tested, but was inferred from the literature;

Lines 131-132 – “These results...environments” should be removed to the Conclusions;

Lines 133-135 – replace “A question...suction” with a subtitle like “Suction-induced burrow excavation in the field”; Replace “One such...figure 3a” with “The density of individuals versus sediment suction was recorded in the intertidal sandflat Isumi”.

Lines 138 – Change “habitat formation” as explained above;

Line 142 – Change “corresponds” with “corresponded”;

Lines 145-150 – This long sentence is not clear;

Line 154 – Change “habitat” with “burrow”;

Line 156 – Change “habitat formation” with “burrowing activity”;

line 159 – “more suitable optimum” sounds odd; you may simply write “above/below G.W.L. OP”;

Lines 161-162 – Please change “formation of habitats” with “excavation of burrows”;

“manifested” is a transitive verb and should be “manifested itself”, better “was observed”;

Lines 164-165 – The statement belongs to the Conclusions;

Lines 166-169 – The sentence is a repetition of the Methods and should be replaced by a subtitle, like “Suction-induced burrowing behaviour following typhon events”;

Line 178 – “manifested” is a transitive verb and should be “manifested itself”, better “was observed”;

Lines 179-181 – The statement belongs to the Conclusions;

Line 181 – Add here a subtitle like “Crab competition for burrows”;

Line 187 – change “numbered” with “The number of thieves was 132”;

Line 190 – Change “created” with “excavated”;

Lines 193-194 – “These findings... society”, see comment for lines 20-21; in any case the statement belongs to the Discussion;

Line 195 – Start here the “Conclusions” section, taking into account the above comments. Please, note that the analysis of burrow excavation and defence is not new; what is new is the orientation with respect to a suction gradient, i.e., a geophysical gradient.

The reviewer: Felicita Scapini, Department of Biology, University of Florence, Italy

Reviewer: 2

Comments to the Author(s)

The findings of this paper show that the sand bubbler crab (SBC) utilizes suction to detect and migrate to suitable environments at different times. These results also show a suction-induced detection, migration and habitat formation by SBC. The authors state that this is a new form of

environmental detection and migration by animals. The capillary suction responds to changes in groundwater levels driven by climate change. These results may also affect conservation strategies.

General comments:

1. English needs review. For example, 'A series of experiments was performed...' Line 115. Change "was" by "were". Review the entire paper for additional edits.
2. Use the acronym SBC in the text as appropriate. There are several sections where the words "sand bubbler crab" could be substituted with SBC. Lines 72, 74, 82, 84, etc.
3. Use past tense to describe aims, methods, and results. For example: "The present study aims to explore...". Change it by "The present study explored the potential of the SBC..." Line 53.

Major comments:

Abstract:

1. What do the authors mean by "better living conditions"? Line 20

Results and Discussions:

1. What is a "suitable suction environment"? Line 132
2. Please explain the logic for the following statement: "a distinct habitat may be formed". Line 154
3. Please explain the statement in line 193-194 "these findings indicate..., as in human society". Are the authors comparing the behavior of SBC with human behavior? If so, what variables of human behavior were compared with SBC behavior?
4. How exactly do the authors suggest that their findings will "broaden the intersection between biology, physics and engineering"?

The findings of this paper are very interesting but could be strengthened by avoiding assumptions rather than the results provided for the experiments and field observations.

Author's Response to Decision Letter for (RSOS-180976.R0)

See Appendix A.

RSOS-190088.R0

Review form: Reviewer 1 (Felicita Scapini)

Is the manuscript scientifically sound in its present form?

Yes

Are the interpretations and conclusions justified by the results?

Yes

Is the language acceptable?

Yes

Is it clear how to access all supporting data?

Yes

Do you have any ethical concerns with this paper?

No

Have you any concerns about statistical analyses in this paper?

I do not feel qualified to assess the statistics

Recommendation?

Accept as is

Comments to the Author(s)

The revised version of the paper has addressed the issues highlighted in my previous review. Rapid publication is recommended.

Still, the following very minor changes would improve the paper:

line 19, "suitable suction environment cause repeated battles", please change with "repeated battles were observed in suitable suction environment";

lines 97-98, the observation period is long and the dates should be reported in the paper; the four seasons are mentioned in figure 3 only and are not four seasons in each site; so "different seasons" would be better;

line 102, please change "created" with a "excavated";

line 117, please change "respond" with "responded";

line 126, please change "movement" with "movements"; "but instead" is pleonastic: please change the expression by using "instead" or "but" only;

lines 160-161: the seasons of the observations were not the same for all sites;

Figure captions at lines 281, 282: please change the word "creating" with "excavating";

Decision letter (RSOS-190088.R0)

24-Apr-2019

Dear Dr Sassa,

I am pleased to inform you that your manuscript entitled "Suction-induced habitat selection in sand bubbler crabs" is now accepted for publication in Royal Society Open Science.

Please ensure that you email the editorial office with an editable file type version (Word or Latex are preferred) of your manuscript, and also individual files for any figures, tables, and captions for figures and tables, as soon as possible. We cannot proceed without these.

You should incorporate the remaining typographical changes recommended during proofing - be careful to include these.

Royal Society Open Science operates under a continuous publication model

(<http://bit.ly/cpFAQ>). Your article will be published straight into the next open issue and this will be the final version of the paper. As such, it can be cited immediately by other researchers. As the issue version of your paper will be the only version to be published I would advise you to check your proofs thoroughly as changes cannot be made once the paper is published.

You have the opportunity to archive your accepted, unbranded manuscript, but access to the full text must be embargoed until publication.

Articles are normally press released. For this to be effective we set an embargo on news coverage corresponding to the publication date of the article. We request that news media and the authors do not publish stories ahead of this embargo (when final version of the article is available).

on behalf of Dr Safi Darden (Associate Editor) and Kevin Padian (Subject Editor)
openscience@royalsociety.org

Reviewer comments to Author:
Reviewer: 1

Comments to the Author(s)

The revised version of the paper has addressed the issues highlighted in my previous review. Rapid publication is recommended.

Still, the following very minor changes would improve the paper:

line 19, "suitable suction environment cause repeated battles", please change with "repeated battles were observed in suitable suction environment";

lines 97-98, the observation period is long and the dates should be reported in the paper; the four seasons are mentioned in figure 3 only and are not four seasons in each site; so "different seasons" would be better;

line 102, please change "created" with a "excavated";

line 117, please change "respond" with "responded";

line 126, please change "movement" with "movements"; "but instead" is pleonastic: please change the expression by using "instead" or "but" only;

lines 160-161: the seasons of the observations were not the same for all sites;

Figure captions at lines 281, 282: please change the word "creating" with "excavating";

Appendix A

The authors' responses to the comments provided by the Editor and the Reviewers

Comments by Editor and the authors' responses:

Comment: This is clearly an interesting and well executed study. However, the reviewers found a great many problems with it, although most were minor and are easily fixed. I am recommending a "reject/resubmit" decision because it will give you more time to revise than a "major revision" decision. Please make sure to address all comments.

Response: We have revised our manuscript by addressing and clarifying all of the comments provided by the reviewers as described below. Please note that all of the revised parts are highlighted in red in the revised manuscript attached herewith.

Comments by Reviewer #1 and the authors' responses:

Comment: The study on the response of sand bubbler crabs to sand-suction dynamics is highly interesting, well conducted and worth of publication. Still, the interpretation of the results and text layout need a thorough revision, as indicated below.

Response: We sincerely appreciate your support for our study. We have revised our manuscript by addressing and clarifying all of the comments provided as described below. Please note that all of the revised parts are highlighted in red in the revised manuscript attached herewith.

Comment: The "detection of sediment suction state" was not analysed, but simply inferred from the observation of behaviour and literature.

Response: We have omitted the description "detection" in the revised manuscript.

Comment: "Migration" was not observed, but the orientation to suction gradient was tested in laboratory experiments, while zonation and zonal recovery after typhon events were analysed in the field.

Response: We have omitted the description "migration" in the revised manuscript.

Comment: "Habitat formation" is a general concept: here burrowing/burrow development/competition for burrows were studied.

Response: We have omitted the description "habitat formation" in the revised manuscript.

Comment: Throughout the text, the terms "environment" and "habitat" are used with different meanings. I suggest to use "beach zone", defined by specific features, including sediment suction; "habitat", when referring to the zone where the SBC live; "burrow", the constructions by the SBCs.

Response: Please note that SBC live not only in sandy beaches but also in sandflats. Accordingly, we used zone when referring to a spatial location, and habitat and burrow as suggested.

Comment: Title – it is misleading, as no detection (of what?) nor migration were observed. I suggest a new title that would better reflect the study: "Suction-induced habitat selection, orientation and burrowing in sand bubbler crabs".

Response: Following the suggestion, we have revised the title as "Suction-induced habitat selection in sand bubbler crabs". Please note that the process of habitat selection involves both orientation and burrowing in addition to movement. Accordingly, we used habitat selection throughout the revised manuscript.

Comment: Please, add the scientific name of the studied species, which is a crustacean decapod. In general the term "invertebrates" should be avoided because it refers to very different animal groups.

Response: We have added the description "decapod crustacean" and the scientific name "*Scopimera globosa*" in the Abstract.

Comment: Please, mention the environment under study, i.e. sand flats, and geographically locate the study area; specify that both laboratory experiments and field observation were carried out.

Response: We have mentioned our study fields that were various sandflats in Japan and specified that both laboratory and field observations were carried out in the Abstract. Please note that the geographical locations of the sandflats are shown in figure S6.

Comment: lines 7, 11: change “environments” with “beach zones”;

Response: We have changed “environments” with “zones” here. Please note that SBC live not only in beaches but also in sandflats, as described above.

Comment: line 8: change “form habitats” with “develop burrows”;

Response: We have omitted the description “form habitats”. Please note that we used habitat selection or select habitats throughout the revised manuscript, on the basis of the reviewer’s suggestion described above.

Comment: explain what “normal times” and “times of events” are;

Response: Normal times denote ordinary times in the absence of disaster events. In order to make this clear, we have revised the description as “at normal times and at the time of disaster events” in the Abstract.

Comment: line 13: it is not clear;

Response: For the purpose of clarity, we have revised the sentence by adding the phrase “the knowledge of”.

Comment: line 14: Please, change “detection, migration and habitat formation” with “Suction-induced habitat selection, orientation and burrowing in sand bubbler crabs”, as suggested above;

Response: We used “Suction-induced habitat selection in sand bubbler crabs”, on the basis of the reviewer’s suggestion, as described above.

Comment: line 18: please change the sentence, as an environment can not “give rise”;

Response: We have changed the description “give rise to” to “cause”.

Comment: line 19: please define “critical suction environment”;

Response: We have defined it as “critical suction environment for burrowing” in the Abstract.

Comment: lines 20-21: Delete the sentence, as this is a naïve statement, which would need a forum discussion, as the drivers of human habitat selection behaviour are much more complicated than crabs’ ones. Sediment suction/soil characteristic as drivers for human constructions are beyond the aims of the study!

Response: We have deleted the sentence.

Comment: Keywords: change “migration” with “orientation”

Response: On the basis of the reviewer’s suggestion described above, we used habitat selection throughout the revised manuscript. Hence, we have changed “migration” to “habitat selection” in Keywords.

Comment: Introduction

Lines 27-28 – This is a naïve statement that ignores the huge literature on orientation to cues of different nature, gradients and directional stimuli, on humans and animals including invertebrates; I suggest to delete the whole sentence and start with the keyword “Capillarity”.

Response: We have deleted the whole sentence and started with the keyword “Capillarity” in the Introduction.

Comment: Lines 31-32 – Please delete “from insects to humans” (no direct relationships, ignoring marine and interstitial animals...); quote a review paper on the subject, which may also include plants, which are biological objects highly influenced by capillarity;

Response: We have deleted the description “from insects to humans” and quoted a review paper here.

Comment: Line 34 – “magical power” is not a scientific concept and should be avoided;

Response: We have omitted the description “magical”.

Comment: Line 39 – change “surficial” with “surface”;

Response: We have changed the phrase as suggested.

Comment: Line 48 – please, add here the scientific name of the species studied; define “critical suction”;

Response: We have added the scientific name “*Scopimera globosa*” and defined the critical suction here with the corresponding reference. The following sentence further explains the definition.

Comment: Line 53-54 – see above

Response: On the basis of the reviewer’s suggestion described above, we used habitat selection or select habitats throughout the revised manuscript.

Comment: Methods

Line 73- change “habitats” with “burrows”;

Response: We have changed the phrase as suggested.

Comment: Lines 77, 92-93 – please geographically locate the study sites;

Response: We have added the geographical location in figure S6.

Comment: Line 82 – How many crabs were tested? At what time of day? What was the treatment before and after the experiments (captive time, feeding, handling, were individuals re-used?);

Response: We tested a total of 623 individuals in our laboratory experiments that were conducted from 1pm to 6pm a day. In all experiments, the air temperature, the water temperature, and the salinity of the water and pore water were kept essentially constant at 20 to 21 °C, 19 to 20 °C, and 27 psu, respectively. Prior to the experiments, SBCs were maintained in the laboratory under aerated fresh seawater in the intertidal sediments for over one month to ensure that any endogenous physiological rhythms were abolished. We did not re-use individuals used for each case. After the experiments, we measured their carapace widths and wet weights. We have added the corresponding descriptions in the Methods section.

Comment: Line 84 – change with “Experiments were carried out to see how burrows are formed by sand bubbler crabs”;

Response: We have changed the sentence as suggested.

Comment: Line 91 – “equal to 10mm”: this is not possible! Kindly write the error or write “around 10mm”;

Response: We have revised the description by including the error such that the carapace widths were 8 ± 1 mm.

Comment: Line 94 – “density surveys”: add here the sampling method and design used, which is specified at lines 166-169;

Response: We have added the sampling method and design in the lines 100-102 in the revised manuscript.

Comment: Line 97 – “in different seasons”: kindly specify the differences between seasons for international readers;

Response: We have specified the differences between seasons in figure. 3 of the revised manuscript. Accordingly, we have revised “different seasons” with “four seasons”.

Comment: “tensiometers”, please specify the characteristics of the instruments used;

Response: We have specified the characteristics of the instrument, together with the corresponding references, such that the device consisted of a plastic tube with a porous ceramic cup, and was filled with deaerated water.

Comment: Lines 110-111 – change “process of migration” with “displacement”; change “creation” with “excavation”;

Response: We have changed them as suggested. The figure caption for figure 1 has also been revised accordingly.

Comment: Results and discussion – The results, discussion and conclusions should be clearly separated. I

suggest to add subtitles instead of repetitions of the methods.

Response: Please kindly note that the journal Royal Society Open Science accepts a combined Results and Discussion section, and the authors would like to follow this type. However, we have separately located a Conclusions section and added subtitles as suggested.

Comment: Lines 115-118 – replace with a subtitle, referring to figure S1, e.g., “Laboratory experiments using the two dimensional flume”;

Response: We have added the subtitle as suggested. Please note that two types of experiments were conducted, and in order to make clear which type of the experiment is referred to, we have kept the description of the experiment type here.

Comment: Lines 120-122 – “Since...suction”: please remove to the Discussion, as this was not tested, but was inferred from the literature;

Response: Please see our response to comments on the section organization above.

Comment: Lines 131-132 – “These results...environments” should be removed to the Conclusions;

Response: We have omitted the corresponding sentence here.

Comment: Lines 133-135 – replace “A question...suction” with a subtitle like “Suction-induced burrow excavation in the field”;

Response: We have added the subtitle “Suction-induced habitat selection in the field”. Please note that on the basis of the reviewer’s suggestion, we used habitat selection throughout the revised manuscript, as described above.

Comment: Replace “One such...figure 3a” with “The density of individuals versus sediment suction was recorded in the intertidal sandflat Isumi”.

Response: We have replaced the sentence as suggested.

Comment: Lines 138 – Change “habitat formation” as explained above;

Response: We have changed the phrase with “habitat selection” as described above.

Comment: Line 142 – Change “corresponds” with “corresponded”;

Response: We have changed it as suggested.

Comment: Lines 145-150 – This long sentence is not clear;

Response: For the purpose of clarity, we have revised the corresponding sentence in lines 149-156 of the revised manuscript.

Comment: Line 154 – Change “habitat” with “burrow”;

Response: Please note that on the basis of the reviewer’s suggestion, we used habitat selection or select habitats throughout the revised manuscript, as described above, and have kept the habitat here since this habitat refers to the zone where the SBC live.

Comment: Line 156 – Change “habitat formation” with “burrowing activity”;

Response: We have changed the phrase with “habitat selection” as described above.

Comment: Line 159 – “more suitable optimum” sounds odd; you may simply write “above/below G.W.L. OP”;

Response: We have revised the phrase as suggested.

Comment: Lines 161-162 – Please change “formation of habitats” with “excavation of burrows”; “manifested” is a transitive verb and should be “manifested itself”, better “was observed”;

Response: On the basis of the reviewer’s suggestion described above, we used habitat selection throughout the revised manuscript. We have also changed “manifested” with “was observed” as suggested in the revised

manuscript.

Comment: Lines 164-165 – The statement belongs to the Conclusions;

Response: We have moved the statement to the Conclusions.

Comment: Lines 166-169 – The sentence is a repetition of the Methods and should be replaced by a subtitle, like “Suction-induced burrowing behaviour following typhon events”;

Response: We have added the subtitle “Suction-induced habitat selection following typhon events”. Please note that on the basis of the reviewer’s suggestion, we used habitat selection throughout the revised manuscript, as described above.

Comment: Line 178 - “manifested” is a transitive verb and should be “manifested itself”, better “was observed”;

Response: We have revised the phrase as suggested.

Comment: Lines 179-181 - The statement belongs to the Conclusions;

Response: In order to avoid the repetition of the statement in Conclusions, we have revised the corresponding sentence in lines 177-179 in the revised manuscript.

Comment: Line 181 – Add here a subtitle like “Crab competition for burrows”;

Response: We have added the subtitle as suggested.

Comment: Line 187 – change “numbered” with “The number of thieves was 132”;

Response: We have changed the phrase as suggested.

Comment: Line 190 – Change “created” with “excavated”;

Response: We have changed the phrase as suggested.

Comment: Lines 193-194 – “These findings... society”, see comment for lines 20-21; in any case the statement belongs to the Discussion;

Response: We have omitted the corresponding sentence.

Comment: Line 195 – Start here the “Conclusions” section, taking into account the above comments. Please, note that the analysis of burrow excavation and defence is not new; what is new is the orientation with respect to a suction gradient, i.e., a geophysical gradient.

Response: We have stated here the Conclusions section. Please note that on the basis of the reviewer’s suggestion, we used habitat selection throughout the revised manuscript, as described above.

Comments by Reviewer #2 and the authors’ responses:

Comment: The findings of this paper show that the sand bubbler crab (SBC) utilizes suction to detect and migrate to suitable environments at different times. These results also show a suction-induced detection, migration and habitat formation by SBC. The authors state that this is a new form of environmental detection and migration by animals. The capillary suction responds to changes in groundwater levels driven by climate change. These results may also affect conservation strategies.

Response: We sincerely appreciate your support for our study. We have revised our manuscript by addressing and clarifying all of the comments provided as described below. Please note that all of the revised parts are highlighted in red in the revised manuscript attached herewith.

Comment: English needs review. For example, ‘A series of experiments was performed...’ Line 115. Change “was” by “were”. Review the entire paper for additional edits.

Response: We have revised the manuscript based on the editing by the English language editor. We have also revised the sentence as suggested.

Comment: Use the acronym SBC in the text as appropriate. There are several sections where the words “sand bubbler crab” could be substituted with SBC. Lines 72, 74, 82, 84, etc.

Response: We have used the acronym SBC as appropriate consistently in the revised manuscript.

Comment: Use past tense to describe aims, methods, and results. For example: "The present study aims to explore...". Change it by "The present study explored the potential of the SBC..." Line 53.

Response: We have used the past tense as suggested.

Comment: Abstract:

1. What do the authors mean by “better living conditions”? Line 20

Response: We have omitted the corresponding sentence in the Abstract.

Comment: Results and Discussions:

1. What is a “suitable suction environment”? Line 132

Response: We have omitted the corresponding sentence here. Please note that the definition is described in lines 44-48 of the Introduction of the revised manuscript.

Comment: Please explain the logic for the following statement: “a distinct habitat may be formed”. Line 154

Response: In order to make the logic clear, we have revised the corresponding descriptions in lines 149-158 of the revised manuscript. Please also refer to the prior statements in lines 132-134 of the revised manuscript.

Comment: 3. Please explain the statement in line 193-194 “these findings indicate..., as in human society”. Are the authors comparing the behavior of SBC with human behavior? If so, what variables of human behavior were compared with SBC behavior?

Response: We have omitted that statement in the revised manuscript.

Comment: How exactly do the authors suggest that their findings will “broaden the intersection between biology, physics and engineering”?

Response: We have revised the sentence such that “this new form of habitat selection by animals will broaden the intersection between ecology and geophysics”. Please note that the ecology here refers to habitat selection and the geophysics does suction-induced geophysical processes in our study.

Comment: The findings of this paper are very interesting but could be strengthened by avoiding assumptions rather than the results provided for the experiments and field observations.

Response: We sincerely appreciate your support for our study. We have revised the whole manuscript on the basis of the results obtained in the present study.